

# The Impact of Geological Structures on Groundwater Potential Assessment in Volcanic Rocks of the Northwestern Ethiopian Plateau: A Review

Bishaw Mihret[1*,] Ajebush Wuletaw[1]

[1]Department of Geology, Debre Markos University, PoBox:269, Debre Markos, Ethiopia.

[*]Corresponding Author Email: bishawmihret2022@gmail.com

**Abstract**

This review examines the influence of geological structures on groundwater potential in the volcanic rocks of the Northwestern Ethiopian Plateau. The region's tectonic complexity has shaped fractures, faults, and other features that significantly impact groundwater storage and flow. Geological structures, including faults, fractures, folds, and lineaments, play a crucial role in groundwater dynamics, particularly in terrains with limited primary porosity, where secondary porosity dominates aquifer characteristics. Faults can act as conduits or barriers, controlling recharge, flow, and discharge based on their structural properties and interaction with surrounding rocks. Fractures create secondary porosity, enabling groundwater storage and movement in otherwise impermeable rocks. Lineaments, representing subsurface features such as faults and lithological boundaries, are key indicators of groundwater potential, especially in hard-rock and volcanic terrains. Additionally, folding influences aquifer configuration and flow by creating confined or unconfined groundwater systems through anticlines, synclines, and other structures. The review underscores the importance of integrating geological, geophysical, and hydrological methods for effective groundwater exploration and management. Volcanic terrains present unique challenges due to their complex lithology and structural heterogeneity. Case studies from various volcanic settings demonstrate how structural features enhance or restrict groundwater movement and highlight the interplay between volcanic lithology and tectonic processes. Recommendations are provided for using a multidisciplinary approach to address these challenges and ensure sustainable groundwater resource management in volcanic regions.

**Keywords:** Geological structures, groundwater potential, volcanic rocks, Ethiopian plateau, hydrogeology

## 1. Introduction

Groundwater is a crucial resource, especially in arid and semi-arid areas where surface water is limited or unreliable (Kebede et al., 2005; Ayenew et al., 2008; Azagegn et al., 2015). In regions with low primary porosity, geological structures like faults, fractures, joints, lineaments, and dykes significantly influence groundwater dynamics. These structures can either act as barriers or conduits for groundwater flow, depending on their characteristics such as orientation, density, connectivity, and permeability (Acocella et al., 2003). Faults and fractures often facilitate groundwater flow, while folds and impermeable layers can obstruct it. The interaction between subsurface fluids and faulting is well-documented (Hardbeck and Hauksson, 1999), making the study of these structures essential for



effective groundwater management, particularly in areas where water resources are scarce. In Ethiopia, groundwater
is vital, particularly in the arid and semi-arid regions where surface water is unreliable. The Northwestern Ethiopian
Plateau, dominated by volcanic rocks formed by Tertiary to Quaternary volcanic activities, is significantly influenced
by tectonic processes, particularly those related to the East African Rift System (WoldeGabriel et al.,1990; Chernet et
al., 1998, Fenta et al., 2020, Tafesse and Alemaw, 2020). This results in a complex array of fractures, faults, and other
geological features that govern groundwater movement. Understanding how geological structures influence
groundwater is essential for managing this resource effectively. This review evaluates the impact of these structures
on groundwater potential in volcanic terrains, focusing on the Northwestern Ethiopian Plateau. Groundwater in
volcanic areas is controlled by the physical properties of volcanic rocks and the structural changes caused by tectonic
activity. Key factors such as lithological heterogeneity, the degree of fracturing, and weathering processes dictate the
distribution of groundwater in these regions (Freeze and Cherry, 1979). In volcanic terrains, faults are particularly
significant. These discontinuities in the Earth's crust can either enhance or restrict groundwater flow, depending on
their displacement, orientation, and associated materials. Faults may serve as conduits for water flow and recharge or
act as barriers to groundwater movement. Therefore, understanding fault dynamics is crucial for groundwater
management, especially in regions with complex geology (Freeze and Cherry, 1979). Volcanic rocks are often
heterogeneous and anisotropic, making groundwater exploration challenging. The movement and storage of
groundwater in these terrains are heavily influenced by geological structures such as faults, fractures, joints, and
lithological contacts. This review aims to provide a deeper understanding of how these structures shape groundwater
potential in volcanic regions, particularly in the Northwestern Ethiopian Plateau (Fig.1) (Ayenew et al., 2008, Nigate
et al. 2020). Fractures, caused by stress in rocks, are essential for groundwater flow in hard-rock terrains. Unlike
primary porosity in sedimentary rocks, fractured rocks rely on secondary porosity to store and transmit groundwater.
This makes understanding the nature and behavior of fractures critical for groundwater exploration in crystalline and
volcanic terrains (Fetter, 2001, Shube et al., 2023). Lineaments, visible as linear features on satellite images, often
indicate zones of structural weakness, such as fractures and faults, that influence groundwater movement. Identifying
and analyzing these lineaments are vital for exploring groundwater resources in areas with complex geological
conditions (Fetter, 2001). Folding, another tectonic process common in volcanic regions, leads to the deformation of
primary lithological units. The resulting folds can affect the orientation, connectivity, and storage capacity of aquifers.
In volcanic terrains, folding has significant hydrogeological implications, as it often leads to the creation of confined
or semi-confined groundwater systems (Tamesgen et al., 2023). The complex interaction between folding and
groundwater movement makes it essential to consider this process when assessing groundwater resources in such
regions. Thus, understanding the role of geological structures in groundwater dynamics is essential for managing water
resources, particularly in volcanic regions like the Northwestern Ethiopian Plateau. Faults, fractures, lineaments, and
folds all play crucial roles in controlling groundwater flow and storage.



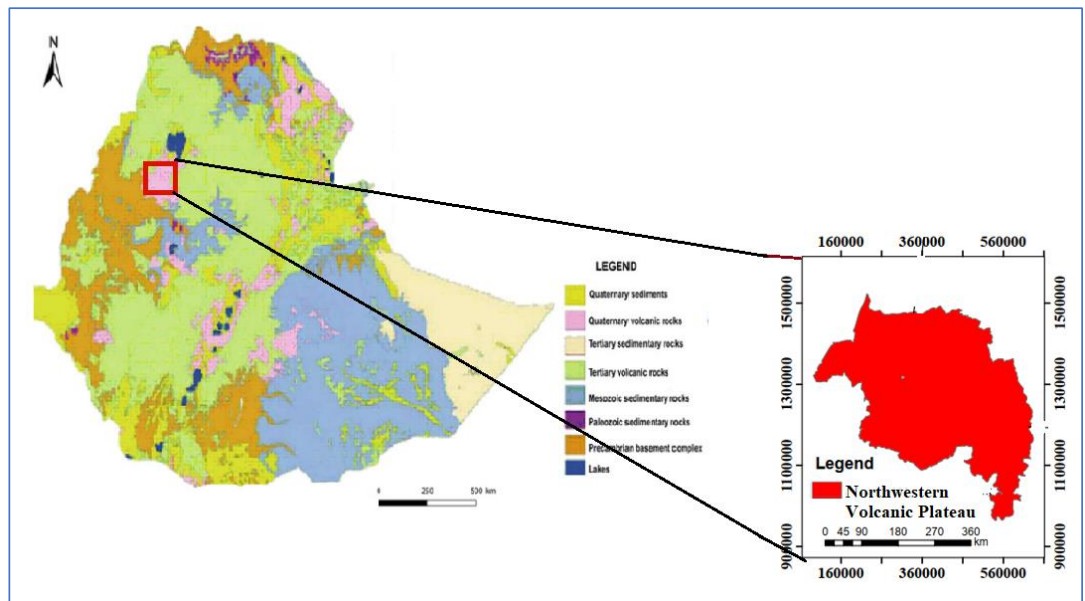

*Figure 1. The spatial distribution of geology by Berhanu et al.,2013 (on the left side) and the study area (Northwestern Volcanic Plateau)*

## 2. Methods for Assessing Structural Influence on Groundwater Potential

Assessing groundwater potential in volcanic terrains requires a multi-faceted approach, integrating geological, geophysical, remote sensing, GIS, and hydrogeological methods.

### 2.1. Geological Mapping

Geological mapping is a crucial tool for understanding the distribution of faults, fractures, and folds in volcanic regions. Detailed structural mapping helps identify key areas for groundwater recharge and defines aquifer boundaries (Mohr and Zanettin, 1988, Abiye, 2020). This method allows for the identification of fault zones, fractures, and variations in rock types critical to groundwater exploration (Kebede, 2013). Field studies are essential for observing surface fractures and correlating them with groundwater potential. Mapping fracture zones helps to assess their orientation, density, and connectivity, which are important for groundwater flow (Fetter, 2001, Kebede et al., 2008). Remote sensing techniques, combined with GIS, enhance lineament detection and analysis. High-resolution satellite images, such as those from Landsat and Sentinel-2, and Digital Elevation Models (DEMs) help identify and analyze lineaments, while GIS tools assist in calculating lineament density, providing valuable information for groundwater mapping (Abiye, 2020).



### 2.2. Geophysical Techniques

Geophysical methods, including electrical resistivity, seismic surveys, and magnetic techniques, are commonly used to explore subsurface structures and aquifers. These methods are effective in detecting fault zones associated with groundwater movement (Fetter, 2001) and in mapping fracture zones within aquifers (Abiye, 2020). Electrical resistivity surveys, in particular, are valuable for high-resolution mapping of shallow fractures, helping to delineate areas with significant groundwater potential (Heath, 1983).

### 2.3. Remote Sensing and GIS

Remote sensing and GIS are powerful tools for lineament mapping and spatial analysis of groundwater potential. By combining remote sensing data with field observations, these tools have improved the efficiency of groundwater exploration (Tesfaye et al., 2020). Satellite imagery, such as from Landsat or Sentinel-2, can be used to map lineaments, revealing fracture patterns that directly correlate with groundwater potential. The integration of GIS allows for spatial analysis that enhances the understanding of groundwater systems and aids in predicting areas of high groundwater yield (Tesfaye et al., 2020).

### 2.4. Hydrogeological Studies

Hydrogeological studies, including aquifer tests, tracer studies, and water table monitoring, are essential for understanding aquifer properties and groundwater movement. These studies provide insights into recharge rates, flow mechanisms, and the dynamics of fractured aquifers. Hydraulic tests, such as pumping and slug tests, help quantify key parameters like hydraulic conductivity and transmissivity in fractured aquifers (Freeze & Cherry, 1979). The results are vital for assessing the productivity of groundwater systems influenced by geological structures. Areas with dense lineament patterns often correlate with high-yield groundwater wells, particularly where lineament intersections occur, as they enhance permeability and groundwater flow (Kebede, 2013). Combining lineament analysis with other hydrogeological data provides a comprehensive understanding of groundwater potential, especially in arid and semi-arid regions, where groundwater is a vital resource (Tesfaye et al., 2020).

### 3. Role of Geological Structures in Groundwater potential

Geological structures are critical in influencing groundwater dynamics in volcanic terrains.

### 3.1. Faults and Their Role in Groundwater Systems

Faults play a significant role in shaping groundwater potential by creating pathways for water flow or acting as barriers. Normal faults often facilitate groundwater recharge, while reverse faults can restrict flow due to compression and low permeability (Freeze & Cherry, 1979). The hydraulic conductivity of fault zones varies depending on the infilling material; materials like clay or gouge reduce permeability, while open fractures enhance it, allowing for easier water movement (Abiye, 2020). In cases where faults are filled with low-permeability materials, such as clay or calcite, they may act as barriers, disrupting groundwater flow and forming perched water tables or isolated groundwater





systems (Abiye, 2020). Faults can enhance groundwater movement in volcanic terrains, particularly where fracturing
and brecciation have occurred. These fractures and fault planes create preferential pathways for water, linking aquifers
and increasing recharge (Fetter, 2001). In volcanic regions, fault zones often correspond with high-yielding wells due
to the secondary porosity they create (Kebede, 2013). Faults are also associated with springs, where groundwater rises
to the surface through fault intersections with aquifers. These springs serve as important indicators of subsurface
hydrogeology and are commonly utilized as drinking water sources in fault-prone areas (Freeze & Cherry, 1979).
However, faults filled with impermeable materials such as clay or silica can reduce permeability and restrict
groundwater flow, making them barriers. The permeability of fault zones is influenced by factors like fault orientation,
the stress field, and the direction of groundwater flow. Vertical faults generally promote vertical water flow, while
horizontal or shallow faults can act as barriers (Fetter, 2001). The width of the fault zone also affects its ability to
facilitate water flow; narrow, well-fractured faults tend to enhance flow, while wider zones filled with gouge material
may impede it (Chernet, 1993). The surrounding lithology further influences fault behavior, with faults in basaltic
rock typically enhancing flow due to the rock's fractured nature, while those in pyroclastic material may have more
variable effects, depending on consolidation and weathering (Kebede, 2013).

**3.2. Fractures and Secondary Porosity**
Fractures play a crucial role in enhancing secondary porosity, which significantly influences groundwater storage and
movement in consolidated rocks. In highly fractured zones, groundwater yields tend to be higher due to increased
permeability and connectivity (Fetter, 2001). In volcanic terrains, for instance, fractured basalts act as primary
aquifers, while unfractured basalts typically serve as aquitards (Kebede, 2013). Fractures allow surface water to
penetrate deeper into the subsurface, enhancing recharge in areas with dense fracturing, which often results in higher
groundwater potential (Chernet, 1993). The effectiveness of fractures as groundwater conduits largely depends on
their connectivity. Well-connected fractures form extensive networks that facilitate both lateral and vertical water
flow, whereas isolated fractures may restrict groundwater movement (Heath, 1983). In hard rocks, like basalt, granite,
and gneiss, groundwater storage is almost entirely dependent on the presence of fractures, as these rocks generally
have low primary porosity (Freeze and Cherry, 1979). The aperture or width of fractures also plays a significant role
in their hydraulic conductivity. Wider fractures allow for greater water flow, while narrow fractures may impede
movement. Fractures infilled with materials such as clay or calcite can reduce hydraulic conductivity and limit water
movement (Fetter, 2001). Additionally, the orientation of fractures relative to the regional stress field and topography
influences groundwater flow. Fractures aligned with the hydraulic gradient promote flow, whereas those oriented
perpendicular to it may hinder movement (Freeze and Cherry, 1979). Higher fracture density is generally associated
with increased groundwater storage and flow, although excessive fracturing can lead to water loss due to rapid
drainage into deeper zones (Abiye, 2020).





### 3.3. Lineaments and Groundwater Potential

Lineaments, which are surface expressions of subsurface geological structures, play a crucial role in groundwater exploration. Studies using remote sensing and GIS have shown that areas with high lineament density tend to have higher groundwater yields (Tesfaye et al., 2020). These linear features often mark zones of increased permeability and recharge potential. Lineaments provide direct pathways for surface water to infiltrate into the subsurface, enhancing recharge in regions where primary porosity is limited. Areas with dense lineaments generally exhibit improved groundwater potential due to the enhanced connectivity between fractures (Chernet, 1993). Lineaments serve as conduits for groundwater flow, particularly in terrains lacking significant primary porosity. Their orientation and connectivity are critical in determining regional groundwater flow patterns (Freeze & Cherry, 1979). In hard-rock and volcanic terrains, lineaments often define areas with increased secondary porosity, which can enhance aquifer storage capacity. These regions are commonly targeted for high-yield wells (Kebede, 2013). The effectiveness of lineaments in influencing groundwater dynamics depends on their depth, width, and the degree of weathering of the underlying rocks (Tesfaye et al., 2020).

### 3.4. Folding and its Impact on Aquifer Systems

Folds, especially anticlines, can create confined aquifers by trapping water between impermeable layers. Synclines, which are trough-like folds with layers dipping towards the center, can serve as groundwater reservoirs when composed of permeable materials like fractured basalts. The impermeable layers at the edges of synclines can prevent lateral water flow, enhancing storage (Freeze & Cherry, 1979). In volcanic terrains, synclines may act as groundwater reservoirs depending on their lithology and structural configuration (Chernet, 1993). Anticlines, arch-like folds where layers dip away from the crest, can trap groundwater beneath impermeable layers, forming confined aquifers that are often under artesian pressure. These aquifers are significant groundwater resources (Fetter, 2001). Recharge zones are typically located along the flanks of anticlines where fractures and faults intersect the surface. Volcanic rocks, with their alternating layers of permeable (e.g., fractured basalt) and impermeable (e.g., volcanic ash) materials, can create complex aquifer systems through folding. Tightly folded volcanic sequences can lead to compartmentalization of groundwater flow, complicating recharge and extraction processes (Kebede, 2013). Folding also generates secondary porosity through fractures formed along fold axes and limbs, which enhances permeability and facilitates groundwater flow. In volcanic terrains, the orientation and density of these fractures are key factors in determining the hydraulic conductivity of folded structures (Heath, 1983).

### 4. Case Studies

### 4.1. Northwestern Ethiopian Plateau

The Northwestern Ethiopian Plateau, part of the larger Ethiopian Highlands, is a significant region for groundwater resources, providing water for both rural and urban populations (Mamo et al. 2020). The plateau features a complex geological setting, with basaltic volcanic rocks, faulting, and sedimentary layers, all of which affect groundwater



availability and movement. This case study examines the geological, hydrological, and environmental factors that
influence groundwater potential in the Northwestern Ethiopian Plateau (Duguma and Duguma, 2022, Asrade, 2024).
Groundwater potential in the volcanic regions of the Northwestern Ethiopian Plateau is significantly influenced by
geological structures and lithology. In this area, fractured basalts and fault zones act as primary aquifers, while
interbedded pyroclastic deposits often serve as aquitards (Kassune et al., 2018). Geophysical surveys and lineament
mapping have been effectively utilized to identify areas with high groundwater yields, contributing to the efficient
management of water resources in the region (Kebede, 2013; Tesfaye et al., 2020). These techniques have proven
particularly useful in locating high-yielding wells, which are often found near major lineaments, highlighting their
critical role in groundwater exploration and development (Tesfaye et al., 2020). The Northwestern Ethiopian Plateau
lies within the Northern Main Ethiopian Rift (NMER) of the East African Rift System (EARS), which trends NE-SW
and connects with the Afar Triple Junction. This region is characterized by active tectonic extension and volcanism
(WoldeGabriel et al., 1990; Chernet et al., 1998). The NMER region also exhibits significant Quaternary faulting and
a complex geomorphological landscape, which further influences groundwater availability (Acocella et al., 2003).
Thus, The Northwestern Ethiopian Plateau has significant groundwater potential due to its unique geological
structures, such as volcanic rocks, fault zones, and sedimentary layers. However, this potential is threatened by over-
extraction, environmental degradation, and climate change. Sustainable groundwater management strategies,
including mapping geological structures, land conservation and reforestation, are essential to ensure the long-term
availability of water for both agricultural and urban needs.

### 4.2. East African Rift System

The East African Rift System (EARS) is one of the most significant geological features in the world, stretching from
the Red Sea in the north to Mozambique in the south. This tectonic plate boundary is characterized by faulting,
volcanic activity, and the formation of deep rift valleys. The geological structures in the EARS such as faults, fractures,
volcanic rocks, and sedimentary deposits play a crucial role in groundwater storage and flow. Understanding the
hydrogeology of the region is essential for assessing the groundwater potential, especially in areas where surface water
resources are scarce or unreliable. A study by Kebede et al. (2021) explored the groundwater potential of the East
African Rift System by examining the hydrogeological properties of the region, including geological mapping,
borehole data, and geophysical surveys. The East African Rift System (EARS) serves as a key example of how tectonic
processes influence groundwater potential in volcanic regions. In this system, faults and fractures enhance secondary
porosity, leading to the development of extensive aquifer systems. However, the complex variability in volcanic
lithology can present challenges in groundwater exploration (Abiye, 2020). Fault zones in the EARS play a crucial
role in groundwater dynamics by acting as recharge pathways, while impermeable volcanic layers limit lateral water
flow (Abiye, 2020). Fractures associated with tectonic activity in the rift are particularly important for groundwater
recharge and storage. Normal faults, along with the fractures they generate, facilitate recharge and support the storage
of water in rift valley aquifers, which is essential for supplying water to arid regions (Abiye, 2020). Additionally,
lineaments formed by faults further enhance recharge and water storage in fractured aquifers, making them critical





sources of groundwater in these drought-prone areas (Abiye, 2020). Folding in volcanic terrains along the EARS
creates alternating layers of permeable and impermeable materials. Recharge primarily occurs along the flanks of
anticlines, while synclinal troughs act as natural storage zones. These folded structures are vital for regional water
supply, especially in arid zones where surface water is scarce (Abiye, 2020). In Ethiopia, groundwater is a major
source of fresh water for domestic, industrial, and agricultural needs, particularly in the absence of reliable surface
water. Ethiopia, often referred to as the "Water Tower of Northeast Africa," is home to numerous rivers that flow from
the highlands to lowland areas and neighboring countries (Alemayehu, 2006). Given the critical role of groundwater,
it is essential to ensure its year-round availability by conducting detailed field investigations, incorporating satellite
imagery, and assessing the region's geological structures and geomorphological features (Srinivasa and Jugran, 2003;
Mondal et al., 2007). Thus, the East African Rift System offers significant groundwater potential due to its complex
geological structures, including volcanic rocks, fault zones, and sedimentary basins. However, this potential varies
greatly across the region, and careful management is required to prevent over-extraction and degradation. Integrated
geological and structural mapping practices, enhanced groundwater recharge, and proper monitoring are essential to
ensure the sustainability of groundwater resources in this critical region.

**5. Challenges and Opportunities**
**5.1. Challenges and Limitations**
Groundwater exploration in the volcanic terrains of the Northwestern Ethiopian Plateau faces several challenges:
**- Data Scarcity:** A major limitation is the lack of high-resolution geological and geophysical data, which hinders a
thorough understanding of the structural controls on groundwater potential. Additionally, the resolution of remote
sensing data may not be sufficient to accurately map lineaments, which are critical for groundwater exploration.
**- Structural Complexity:** The variation in fault orientations, fracture densities, and lithological diversity complicates
the prediction of groundwater flow paths. The anisotropic nature of fractured and folded aquifers further complicates
flow modeling and groundwater movement predictions.
**- Climate Variability:** Unpredictable rainfall patterns impact recharge rates and groundwater availability. Changes
in precipitation due to climate fluctuations affect the reliability of structurally controlled aquifers, especially in regions
with complex geological structures. Variations in recharge rates can undermine the consistency of groundwater
resources, especially in folded aquifer systems where recharge mechanisms are less predictable.
- **Complex Flow Paths:** In volcanic regions, groundwater movement often follows intricate and unpredictable flow
paths, exacerbating difficulties in estimating groundwater availability and potential. The interactions between
structural features, such as faults and fractures, with surface and subsurface conditions are not easily modeled.



### 5.2. Opportunities

**- Advanced Mapping Techniques:** Remote sensing and Geographic Information Systems (GIS) offer valuable tools for mapping and characterizing geological structures like folds, faults, and fractures in volcanic terrains. These technologies enable more accurate identification of groundwater recharge zones and flow pathways. Furthermore, advancements in geophysical techniques, such as electrical resistivity and seismic surveys, allow for better mapping of fault zones and aquifer systems.

**- Integrated Approaches:** Combining geological, geophysical, and hydrogeological data is a promising strategy for improving groundwater management, especially in complex volcanic regions. Integrated approaches allow for a more comprehensive understanding of the dynamics of fault-controlled aquifers and fractured groundwater systems. By synthesizing multiple datasets, more accurate predictions of groundwater availability and sustainable management strategies can be developed.

**- Innovative Tools and Algorithms:** The use of advanced algorithms to automate the detection and analysis of lineaments and other geological structures can significantly enhance the accuracy and efficiency of groundwater exploration. These innovations also allow for improved mapping of fracture-controlled aquifers, which are critical in volcanic terrains where primary porosity is often absent.

### 6. Conclusion

Geological structures are fundamental in determining groundwater dynamics in the volcanic rocks of the Northwestern Ethiopian Plateau. This review synthesizes existing research, emphasizing the critical role of faults, fractures, and lithological variations in groundwater potential assessments. The integration of advanced techniques and addressing data gaps will be vital for ensuring sustainable groundwater resource management in the region. Faults have a dual impact on groundwater potential, acting both as conduits and barriers, depending on their structural features and the materials that fill them. A comprehensive understanding of the hydrogeological behavior of faults is essential for effective groundwater exploration and management. Advances in mapping technologies, geophysics, and remote sensing are increasingly enhancing our ability to assess fault-controlled aquifers and develop sustainable groundwater systems. Fractures are a key component in groundwater systems, particularly in hard-rock and volcanic terrains where primary porosity is often minimal. Their effectiveness as groundwater conduits and storage zones is determined by factors such as orientation, density, and connectivity. Advances in geophysical methods, remote sensing, and hydrogeological studies have significantly improved our understanding of fracture-controlled aquifers, which are vital in many volcanic regions. Lineaments are crucial for exploring groundwater systems, particularly in areas with low primary porosity. These structural features serve as conduits for recharge and groundwater flow, making them prime targets for high-yielding wells and sustainable water resource management. The development of remote sensing, GIS, and geophysical tools has greatly enhanced lineament analysis, providing new opportunities for groundwater exploration in complex geological environments. Folding, particularly in volcanic rocks, significantly impacts aquifer systems by influencing groundwater storage, flow, and recharge. Anticlines and synclines, along with their associated



fractures, shape groundwater dynamics, making an understanding of folded volcanic terrains essential for effective
exploration. The complexity of these folded systems highlights the importance of integrating structural and lithological
data for successful groundwater management. Thus, by integrating multidisciplinary approaches—combining
geology, geophysics, hydrogeology, remote sensing, and GIS—is crucial for improving groundwater resource
management in the volcanic terrains of the Northwestern Ethiopian Plateau and similar regions. Addressing current
challenges and leveraging new technologies will enable the development of sustainable groundwater resources to meet
the needs of growing populations in such areas.

**7. Recommendations and Future Directions**

To enhance groundwater potential assessment in the Northwestern Ethiopian Plateau, the following steps are
recommended:
**1. Integrated Approaches:** Combining geological, geophysical, and hydrological techniques for comprehensive
groundwater assessments is crucial. A multidisciplinary approach will provide a more holistic understanding of the
region's groundwater systems and improve the accuracy of potential zones identification.
**2. High-Resolution Mapping:** The use of advanced remote sensing and GIS technologies is essential for improving
the identification of groundwater potential zones. High-resolution imagery, coupled with GIS tools, will help delineate
fault zones, fractures, and other structural features that influence groundwater availability, leading to more accurate
and efficient exploration efforts.
**3. Long-Term Monitoring:** Establishing monitoring networks across key regions will allow for the ongoing
assessment of groundwater systems, particularly to track the impact of climatic fluctuations and structural changes on
groundwater recharge and flow patterns. Long-term data will help in predicting future groundwater trends and guide
sustainable resource management.
**4. Develop Robust Models:** Future research should focus on developing advanced models that integrate structural
geology, hydrological, and climatic data. These models would provide a dynamic and predictive understanding of
groundwater systems, enabling more effective and sustainable groundwater management. Simulating various
scenarios, such as climate change or land-use modifications, will be essential for ensuring the long-term viability of
groundwater resources in volcanic terrains.
**Funding**
The authors declare that no specific funding was received for conducting this study.
**Data Availability Statement**
The data supporting the findings of this study are provided within the manuscript.
**Competing Interests Declaration**
The authors declare that they have no competing interests.





**Author Contributions**
**Bishaw Mihret:** conceptualized the study, designed the methodology, experimented, and performed data analysis.
**Ajebush Wuletaw:** contributed to writing the manuscript, provided supervision, reviewed the manuscript, and
contributed to critical revisions. All authors read and approved the final manuscript.

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
