# Peer review of "The Impact of Geological Structures on Groundwater Potential Assessment"

_EGUsphere, 2024_

## Referee Comment (RC1)

Referee comment (egusphere-2024-4201)

Ms. Title: The Impact of Geological Structures on Groundwater Potential Assessment in Volcanic Rocks of the Northwestern Ethiopian Plateau: A Review

This manuscript provides a comprehensive review of the influence of geological structures on groundwater potential in the volcanic rocks of the Northwestern Ethiopian Plateau. The authors effectively synthesize existing research, highlighting the critical role of faults, fractures, lineaments, and folds in shaping groundwater dynamics. The review is well-structured, with a clear focus on the interplay between tectonic processes, lithological heterogeneity, and groundwater movement. The incorporation of case studies from the Northwestern Ethiopian Plateau and the East African Rift System adds depth to the discussion, illustrating the practical implications of the reviewed concepts. The manuscript is well-written, with a logical flow and appropriate use of references to support key points. However, there are areas where the manuscript could be improved, particularly in terms of clarity, depth of analysis, and addressing potential limitations.

**Specific Comments on each sections**

**# Abstract and Introduction:**

The abstract provides a good overview of the review's scope and key findings. However, it could be more concise and focused. For instance, the sentence "The review underscores the importance of integrating geological, geophysical, and hydrological methods for effective groundwater exploration and management" could be rephrased to emphasize the novelty or specific contributions of the review.

The introduction effectively sets the stage for the review by highlighting the importance of groundwater in arid and semi-arid regions and the role of geological structures in groundwater dynamics. However, how it could benefits and advances existing knowledge?

**# Methods for Assessing Structural Influence on Groundwater Potential:**

This section talks several methods, including geological mapping, geophysical surveys, remote sensing, and hydrogeological studies. However, the discussion could be enhanced by providing more specific examples or case studies where these methods have been successfully applied in the Northwestern Ethiopian Plateau or similar regions.

The authors mention the use of remote sensing and GIS for lineament mapping but could elaborate on the limitations of these techniques, particularly in regions with complex geology or limited data availability.

**Role of Geological Structures in Groundwater Potential**

The discussion on faults, fractures, lineaments, and folds is thorough and well-supported by references. However, the authors could provide more critical analysis of the variability in the impact of these structures on groundwater potential. For example, while faults can act as conduits or barriers, the conditions under which they favor one role over the other could be explored in greater detail.

**Case Studies**

The case studies for the Northwestern Ethiopian Plateau and the East African Rift System are informative and relevant. However, the authors could provide more detailed analysis of the specific geological and hydrological conditions in these regions, particularly how they influence groundwater potential. For instance, the discussion on the East African Rift System could include more information on the variability in groundwater potential across different segments of the rift by associating with clear structural influence by reviewing different works.

The authors mention the challenges of over-extraction and environmental degradation but could provide more specific examples or data to support these claims.

**Challenges and Opportunities:**

This section focuses on challenges and limitations is well-articulated, particularly the discussion on data scarcity and structural complexity. However, the authors could provide more specific recommendations for addressing these challenges, such as the use of advanced geophysical techniques or the integration of machine learning algorithms for data analysis.

The opportunities section is promising but could be expanded to include more specific examples of how advanced mapping techniques and integrated approaches have been successfully applied in similar regions.

**Conclusion and Recommendations:**

The conclusion effectively summarizes the key findings of the review and highlights the importance of integrating multidisciplinary approaches for groundwater management. However, the authors could provide more specific recommendations for future research, particularly in terms of addressing data gaps and developing predictive models.

**Technical Corrections:**

Line 8-9: "This review examines the influence of geological structures on groundwater potential in the volcanic rocks of the Northwestern Ethiopian Plateau." Consider rephrasing for clarity: "This review examines how geological structures influence groundwater potential in the volcanic rocks of the Northwestern Ethiopian Plateau."

---

## Community Comment (CC1)

Title - **Impact of Geological Structures on Groundwater Potential Assessment in Volcanic Rocks of the North western Ethiopian Plateau: A Review**

**Manuscript Number-** egusphere-2024-4201

**Comments to the Author**

The manuscript presents the review geological Structure and the petrographic volcano rocks in the Northwestern Ethiopian Plateau to understanding the impact on ground water potential assessment. This view of reviewing is important for scientific implication to hydrology and hydrogeological discipline to know about more the groundwater system in volcanic geology. Geological structures are crucial for groundwater flow system in act as permeability (interconnected pores spaces). The present topic of the manuscript is within the aims and scopes of the journal. However, the manuscript addresses the following comments before acceptance of publication like: -

1. the paper deals about the regional discussion and broad way, no clear study area boundary, so please prepare the location map and with clear geographic coordinates, some sentences are grammatically incorrect.

Therefore, I recommended the manuscript for minor revision before publication in Journal of Hydrology and Earth System Sciences (HESS).

---

## Community Comment (CC2)

**Response to the Referee**

Dear referee thanks you for your valuable feeds these makes my paper more strengthen to scientific importance.

Yes, the paper deals more regional way in Northwestern Plateau of Ethiopia,

So, we have made revision of our manuscript based your comments and journal standards.

Like the title: **The Impact of Geological Structures on Groundwater Potential Assessment in Volcanic Rocks in the Borena Saynit district, Northwestern Ethiopian Plateau: A Review**.

We prepared the **Study Area and Location Map**: Provide a clear map that defines the study area with precise geographic coordinates (see below). So the study area is located in Borena saynit district, Southwollo zone, Northwestern Plateau.

[Figure]

*Figure 1. The location map of the study area*

Thank your consideration, reviewing and valuable comments.

We hope, we look forward your positive response to our work.

Best of regards,

*Corresponding Authors

---

## Community Comment (CC3)

**Response to the Referee**

Dear referee thanks you for your valuable feeds these makes my paper more strengthen to scientific importance.

Yes, the paper deals more regional way in Northwestern Plateau of Ethiopia,

So, we have made major revision of our manuscript based your comments and journal standards.

Like the title: **The Impact of Geological Structures on Groundwater Potential Assessment in Volcanic Rocks in the Borena Saynit district, Northwestern Ethiopian Plateau: A Review**.

We prepared the **study area and location map** which provide a clear map that defines the study area with precise geographic coordinates (see below). So, the study area is located in Borena saynit district, Northwestern Plateau. And Structural map (e.g. lineament map) that significant to infer the orientation of structures and groundwater flow system, etc. Please see the revised manuscript.

[Figure]

*Figure 1. The location map of the study area*

Thank your consideration, reviewing and valuable comments.

We hope, we look forward your positive response to our work.

Best of regards,

*Corresponding Authors

---

## Author Comment (AC2)

Dear Reviewer,

Thank you for your valuable feedbacks, those feedbacks and suggestions are very important to strengthen my paper quality.

I correct all comments one by one on the revised manuscript based the journal standard.

| General Comments | I try to correct into consistent and surprising findings based on the review paper and journal standards. This strength my paper quality. |
| --- | --- |
| Specific Comments: | I accept your comments, I try to correct them especially referencing and the idea correlation, sentences bias. I can correct on the revised manuscript. |

I hope this finds you well. I look forward your positive response.

With kind regards

Bishaw Mihret

*corresponding author.

---

## Author Response (AR1)

Dear Reviewer,

Thank you for your valuable feedbacks, those feedbacks and suggestions are very important to strengthen my paper quality.

I correct all comments one by one on the revised manuscript based the journal standard.

| Abstract and Introduction | Thank you, your comments and suggestion, I try to correct by rewrite and elaborate more. |
|---|---|
| Methods for Assessing Structural Influence on Groundwater Potential | Yes, the methodology is somewhat complex and inconsistent incase of reviewing papers on this section |
| Role of Geological Structures in Groundwater Potential | I put the detail structural features with evidences of field photographs and digitizing in map on the revised manuscript |
| Case Studies | I accept your suggestions and comments, so I address specific case studies based the objectives my reviews. |
| Challenges and Opportunities | I correct on the revised manuscript paper, and put specific methods and ways of opportunities to the review |
| Conclusion and Recommendations | Yes, try to rewrite again the concept based on the objectives of the research review guideline and your suggestions. |
| Technical Corrections | I correct it on revised part like "This review examines how geological structures influence groundwater potential in the volcanic rocks of the Northwestern Ethiopian Plateau." |

With kind regards,

Bishaw Mihret

*corresponding author.

**Response to the Referee**

Dear referee thanks you for your valuable feedbacks these makes my paper more strengthen to scientific importance.

Yes, the paper deals more regional way in Northwestern Plateau of Ethiopia, and you raised more detail suggestions and feedbacks

So, we have made major revision of our manuscript based your comments and journal standards.

Like the title: **The Impact of Geological Structures on Groundwater Potential Assessment in Volcanic Rocks in the Borena Saynit district, Northwestern Ethiopian Plateau: A Review.**

We prepared the **study area and location map** which provide a clear map that defines the study area with precise geographic coordinates (see below). So, the study area is located in Borena saynit district, Northwestern Plateau. And Structural map (e.g. lineament map) that significant to infer the orientation of structures and groundwater flow system, etc.

We modify the regional Regional and Hydrogeological setting of the review study area

We wrote clear finding on case study part with field photographs of the review region

Please see the revised manuscript.

[Figure]

*Figure 1. The location map of the study area*

[Figure]

*Figure 2. The lineament map of the study area (Borena Saynit district)*

Thank your consideration, reviewing and valuable comments.

We hope, we look forward your positive response to our work.

Best of regards,

*Corresponding Authors